# Crossroads of Continents and Modern Boundaries: An Introduction to Inuit and Chukchi Experiences in the Bering Strait, Beaufort Sea, and Baffin Bay

**Henry P. Huntington** [1,*], **Richard Binder Sr.** [2], **Robert Comeau** [3], **Lene Kielsen Holm** [4], **Vera Metcalf** [5], **Toku Oshima** [6], **Carla SimsKayotuk** [7] **and Eduard Zdor** [8]

1    Ocean Conservancy, Eagle River, AK 99577, USA
2    Inuvik, NT X0E 0T0, Canada; richardbindersr@gmail.com
3    Iqaluit, NU X0A 0H0, Canada; comeauu@gmail.com
4    Greenland Institute of Natural Resources, Nuuk 3900, Greenland; leho@natur.gl
5    Eskimo Walrus Commission, Nome, AK 99762, USA; vmetcalf@kawerak.org
6    Qaanaaq 3971, Greenland; oshimatoku@gmail.com
7    North Slope Borough Department of Wildlife Management, Kaktovik, AK 99747, USA; Carla.Kayotuk@north-slope.org
8    Department of Anthropology, University of Alaska Fairbanks, Fairbanks, AK 99775, USA; eduard.zdor@gmail.com
*    Correspondence: henryphuntington@gmail.com

**Abstract:** The homeland of Inuit extends from Asia and the Bering Sea to Greenland and the Atlantic Ocean. Inuit and their Chukchi neighbors have always been highly mobile, but the imposition of three international borders in the region constrained travel, trade, hunting, and resource stewardship among neighboring groups. Colonization, assimilation, and enforcement of national laws further separated those even from the same family. In recent decades, Inuit and Chukchi have re-established many ties across those boundaries, making it easier to travel and trade with one another and to create new institutions of environmental management. To introduce Indigenous perspectives into the discussion of transboundary maritime water connections in the Arctic, this paper presents personal descriptions of what those connections mean to people who live and work along and across each of the national frontiers within the region: Russia–U.S., U.S.–Canada, and Canada–Greenland. Some of these connections have been made in cooperation with national governments, some in the absence of government activity, and some despite opposition from national governments. In all cases, the shared culture of the region has provided a common foundation for a shared vision and commitment to cooperation and the resumption of Indigenous self-determination within their homelands.

**Keywords:** Inuit; Chukchi; Arctic; maritime waters; sovereignty; mobility; wildlife

## 1. Introduction

"Water connects us."

Nina Keukei, Ankalin Chukchi, after dancing to the Bering Land Bridge Song performed by George Noongwook, St. Lawrence Island Yupik, at an Arctic event in Washington, DC, September 2019.

Inuit are predominantly a maritime people whose homeland reaches from the Asian coast of the Bering and Chukchi seas across the northern waters of North America to the Atlantic shores of eastern Greenland [1,2]. Some of their Asian neighbors, the Chukchi, also hunt and fish in the sea, extending the boundaries of shared cultural practices (Figure 1). From time immemorial, Inuit and Chukchi have traveled on water and ice throughout this region, to hunt, fish, socialize, and trade [3].

Their interactions have included neighboring peoples such as Koryak, Athabascan, Aleut, Dene, Cree, and Innu. Inuit themselves comprise several distinctive language groups, including Yup'ik, Cupik, Siberian Yupik, Iñupiaq, Inuvialuit, Inuit, Inughuit, Kalaallit, Tunumiut, and others [4]. These groups share a common language family as well as maritime hunting practices and a high degree of mobility on land and sea. The Chukchi include maritime hunters as well as inland reindeer herders, an unusual combination in the Arctic part of Russia [5].

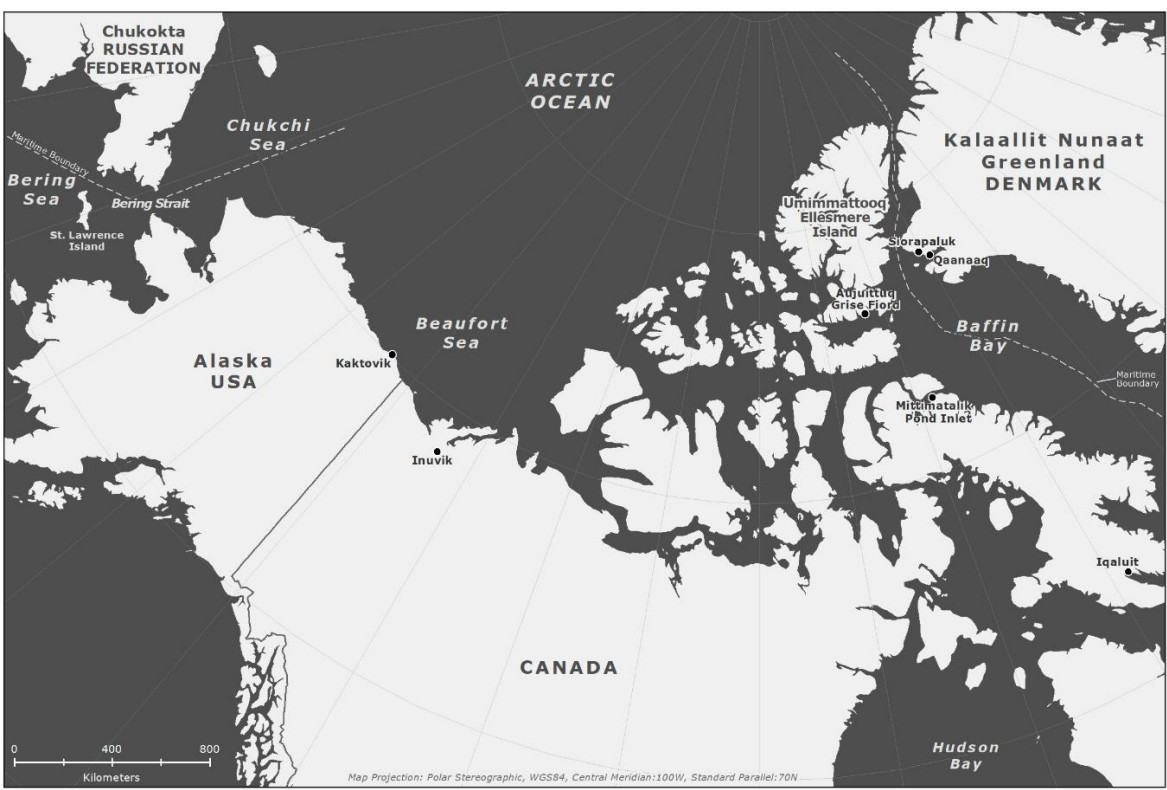

**Figure 1.** Map of the region and places mentioned in the text.

Historically, the arrival of European peoples in the Inuit and Chukchi homelands initially provided new trade opportunities [6], but competition among national powers soon resulted in territorial claims to exclude competitors. Greenland became a colony of Denmark in the 1700s, achieving self-rule status within the Kingdom of Denmark in 2009 [7]. What is now northern Canada was claimed by the British Empire, driven by fur trade, the quest for the Northwest Passage, among other ambitions [8]. Russia took effective control of Chukotka and Alaska in the 1700s [9]. The United States purchased Alaska from Russia in 1867, with no discussion with and minimal awareness of the various Indigenous peoples whose homeland it was and is [10]. In Alaska and Canada, land claims settlements and other legal measures have included recognition of Indigenous identity and rights, which take different forms in the two countries [10]. Greenland makes no such legal distinction, though nearly 90% of its residents were born there [11] and a large majority are of Indigenous descent. Russia recognizes Indigenous identities but rights are another matter [9]. In all four countries, Indigenous persons are subject to the same citizenship rules and procedures as other persons.

Indigenous mobility was not affected at first by the imposition of territorial claims, but the 20th century saw the hardening of borders and the imposition of distinctions of citizenship depending on which side of those borders one's community was associated with. These actions were taken by national governments despite international agreements such as the Jay Treaty of 1794 [12], which promised free travel to Indigenous peoples on either side of the U.S.-Canada border, and the United Nations Declaration on the Rights of Indigenous Peoples [13], which states in Article 36:

Indigenous peoples, in particular those divided by international borders, have the right to maintain and develop contacts, relations and cooperation, including activities for spiritual, cultural, political, economic, and social purposes, with their own members as well as other peoples across borders.

The creation of the Soviet Union in particular first limited trade across the Bering Strait, and then the Cold War saw the imposition of the Ice Curtain in the same region, separating families and friends [9,14,15]. The boundaries between the U.S. and Canada and between Canada and Greenland remained relatively porous for longer, but eventually hunting and other rights were limited to citizens of the respective countries, effectively barring many aspects of mobility, a trend that continues today.

People are not the only species in the Arctic using waters to move and find food. Many of the marine mammals that sustain Inuit and the seabirds that provide fresh meat and eggs are highly migratory, also crossing today's national borders. Limiting access to these species was one of the original aims of territorial claims in the Arctic, though such limitations did little to stop the near extermination of species such as the bowhead whale and walrus [16].

Today, Inuit and Chukchi have led the way in re-establishing ties across the waters of these externally imposed borders, resuming interactions that existed for generations and promoting shared stewardship of shared resources. At the same time, national governments are imposing some stricter controls on international travel in the Arctic, as described below. We are unaware of any previous attempt to document Indigenous transboundary experiences across this region. An introductory explanation of those experiences in light of the tensions between Indigenous-led efforts to restore ties and government efforts to exert control therefore calls attention to and starts to fill a notable gap in the available literature. Studies of Indigenous transboundary experiences elsewhere with water and other topics [17,18] suggest that Indigenous rights and needs are often ignored or marginalized (though not always [19]), their stories lost amid dominant narratives that focus on state actions, international relations, and economics [17].

In this paper, Indigenous authors describe their own experiences with the three international borders within the region: Russia–U.S., U.S.–Canada, and Canada–Greenland. Providing information in the form of stories is a characteristic Indigenous approach [20,21], with individual stories contributing to "a collective story in which every indigenous person has a place" ([22]:144). Some experiences reflect the joy of re-establishing family and cultural connections. Others illustrate the pain of continued separation of people from relatives and territories. Each pair of authors includes one from each side of the border in question and also a man and a woman to provide gender equity. Each contributor has chosen one aspect of transboundary experience to describe. The rest of the paper was drafted by the lead author, who is of European descent, was raised in the eastern U.S., and has lived in Alaska and worked with Arctic peoples for over three decades. He invited each of the rest of the authors to contribute, based on their transboundary experiences. A brief series of first-hand accounts cannot claim to be comprehensive, but it does provide insight into the lived experience of maritime boundaries through the voices of those who are directly affected, day in and day out. This introduction to transboundary maritime connections in the Arctic suggests further emphasis on Indigenous experiences, as explored in the Discussion, and a need to provide places for sharing of additional stories to record the collective experience of those living across the region.

## 2. The Bering Strait

The Bering Strait region includes Chukchi, Iñupiaq, and Siberian or St. Lawrence Island Yupik, and is split by the Russia–U.S. border. Vera Metcalf, St. Lawrence Island Yupik, is a long-time practitioner of co-management for walrus in Alaska and international cooperation across the Bering Strait. Eduard Zdor is a Chukchi scholar and activist currently studying at the University of Alaska Fairbanks and has also worked extensively in wildlife management.

### 2.1. Payangituq, by Vera Metcalf

During the 1920s to 1940s, cultural exchanges continued between Alaska, part of the U.S., and Chukotka, in the Russian Far East. After World War II ended, however, visitations ceased and the international border closed for nearly 50 years. The closure (known as the Ice Curtain) disrupted the freedom for inhabitants on both sides to meet freely with friends and relatives despite kinship and family ties. Despite the closure, people on St. Lawrence Island, on the Alaska side of the border, continued to keep their connection to Chukotka knowing the ties of clanship, family relationship, and knowledge of the Yupik language and *atuq* (traditional drumming, singing, and dancing).

When cultural exchanges resumed after a 1989 agreement was signed between Russia and the U.S. breaking the 50+ year Ice Curtain and creating a visa-free travel program for residents close to the boundary, it was an emotional and life-changing event for St. Lawrence Island Yupiks. The door was open to discovering personal connections and inspiring a revival of our shared culture, language, and song. I am an example of those ties since my clan, the Qiwaaghmiit, originated in a village in Chukotka, called Qiwaa.

The significance of this close connection of people across borders extends to all of our natural resources. We share the Bering Strait, which is a migration route for vast numbers of marine mammals—bowhead and other whales, walrus, all types of seals—and millions of seabirds and fish, too. This is an immense migration that follows the sea ice retreat in spring and its expansion in the fall. Our natural resources are theirs, their natural environment is ours; and we both depend on these for our well-being, nutritionally, economically, and culturally. The strength of our connection to each other, even during times when governments separate us, will hopefully always remain. That spirit is embodied in the word our Chukotkan relatives spoke when they met us again, a word announcing visitors and greeting them and welcoming them: *Payangituq*.

### 2.2. A Water World, by Eduard Zdor

I love the prose of the Chukchi writer Yuri Rytkheu and reread it in moments of joy and adversity. His seemingly unpretentious novels and stories are filled with passions and wisdom of the peoples who settled on the Bering Strait shores several millennia ago. The first time I read Rytkheu as a teenager. His writing led me to a vague understanding that the Bering Strait is not a water border separating the two countries, but a water world that provided a life for peoples, which one of my fellow countrymen very romantically and accurately called Arctic Civilization.

Perhaps that is why I willingly helped in the Bering Bridge Expedition in 1989. The Bering Strait Indigenous residents, including my friends Vadim and Nikolai, took part in this expedition by dog team with Indigenous participants from Chukotka and Alaska. For me, this expedition was a kind of symbolic beginning of the reunification of peoples, separated by the political map of the world. Later, I joined in the work of the indigenous leaders of both sides of the Bering Strait. Vladimir Etylin, Charlie Johnson, Ludmila Ainana, Peter Typykhkak, George Ahmaogak, Vera Metcalf, Gennady Inankeuyas, Igor Makotrik, Harry Brower, and many other activists who contributed to the joint management of wildlife, the reunion of disconnected relatives, and the preservation of native cultures. Behind these official statements is the essence of the work that contributes to the preservation of our peoples' identities. Because the Bering Strait does more than provide traditional food, it sets the rhythm of life, supporting our traditions and spirituality.

Many years ago, I heard a speech by John Waghiyi from St. Lawrence Island at the Beringia Days conference. He is perhaps one of the most active Indigenous leaders seeking to maintain the family ties of the Siberian Yupik. That day he said, "Traditions help us survive. Keeping traditions, we maintain the life path that our ancestors passed on to us and we pass it on to our children." His speech, like Rytkheu's prose, despite the obviousness of the thoughts, sounded to me like a program for the survival of the Bering Strait Region Indigenous peoples. The Bering Strait is not a border, but a unique habitat of the Chukchi, Inuit, and Siberian Yupik, which provides a chance to preserve their cultures, languages, and identity.

### 3. The Beaufort Sea

The Beaufort Sea coast is home to Iñupiat in Alaska and their close relatives the Inuvialuit in Canada. The Canada-U.S. maritime border remains in dispute between the two countries. Carla SimsKayotuk, Iñupiaq from Kaktovik, is a wildlife manager whose husband was born on the Canadian side of the border. Richard Binder Sr., Inuvialuq from Inuvik, Northwest Territories, Canada, is retired from a long career working in environmental management through the co-management organizations created by the Inuvialuit Final Agreement of 1984.

#### 3.1. Paperwork, by Carla SimsKayotuk

In the 1930s and 1940s, many people traveled back and forth along the northern coast of Alaska and Canada, looking for work or better trapping or for other reasons. So many people in the area have parents who were born on one side while they themselves were born on the other. At that time, there was no problem going back and forth, which families did often along the coast in summer and in winter. Today it is a different story, at least for those whose official papers do not match where they are living now. Some were born in Canada, so they have a Canadian passport, but live in Alaska. At present, crossing the border by car on the Alaska Highway does not require a passport, so identification and a U.S. Bureau of Indian Affairs card certifying tribal status are sufficient. Air travel, however, requires a passport. Unless they have official residency status in the U.S., they may be prevented from coming back into the U.S. Oddly enough, many people in this situation are recognized as tribal members in the U.S., even if they are not citizens. So some government agencies accept their status in the U.S., but other agencies do not. All of this is despite the Jay Treaty of 1794, which recognized the right of Native Americans along the border to travel freely between the U.S. and Canada. This will get worse in 2020 when the U.S. requires "Real ID," for which proof of citizenship is required. So those with ambiguous status will have to find a way to clear things up. Otherwise, families will be divided, including brothers and sisters, parents and children, and husbands and wives. Our blood, our songs, our language, our coasts, our waters, and our animals all connect us. But paperwork divides us.

#### 3.2. One People, One Ecosystem, by Richard Binder Sr.

Our people used to go back and forth across what is now the Canada–U.S. border, visiting relatives or following the animals and the seasons. The animals, too, go back and forth each year. But now there is a border dividing us. This creates hardship, because we cannot trade with our relatives for foods we like, such as black maktak (the skin and blubber of the bowhead whale). The whales swim from Alaska to Canada and back, but we cannot transport a part of the whale on the same route. Recently, there have been some accommodations by government authorities that allow us to transport traditional cultural items, such as clothing made of marine mammal skins, so that is good, but there is still a ways to go. In the 1980s, we Inuvialuit from Canada got together with our Iñupiaq cousins from Alaska's North Slope to create a shared, Indigenous-to-Indigenous system for managing the polar bear population that spans our lands and waters on both sides of the border. We were doing what the Canadian and U.S. federal governments were unable to do: cooperate to take care of what is important. We have expanded that cooperation to other species, too. It is good that we can work together with the Iñupiat of Alaska, and that the governments on both sides have come to recognize the importance of our work. But it is also frustrating because the international boundary makes it harder than it should be, with extra rules and complications all because of a line on a map. We are one people and this is one ecosystem, which we should be taking care of together as our ancestors did.

### 4. Baffin Bay

Baffin Bay lies between Kalaallit Nunaat, the Indigenous name for Greenland, and Nunavut, the Canadian territory established in 1999. Along its shores live Inuit in Canada and Kalaallit in Greenland, the northernmost of whom call themselves Inughuit. Toku Oshima, Inughuaq (singular of

Inughuit), is a community leader, hunter, and electrician from Qaanaaq, in northern Greenland. Lene Kielsen Holm, Kalaaleq (singular of Kalaallit), originally from Qaqortoq and living in Nuuk for the past 35 years, helped Toku. Robert Comeau, Inuk (singular of Inuit), is currently a law student in Iqaluit, the capital of Nunavut, Canada.

### 4.1. Umimmattooq and the Qaanaarmiut, by Toku Oshima

I used to go hunting on Umimmattooq (the Greenlandic name for Ellesmere Island) but now I cannot because the border has been closed by the Canadian government. When I was little, there were many people who went with their dog teams to Umimmattooq from Qaanaaq, to go hunting or to visit friends and family in Aujuittuq (the Inuktitut name for the community of Grise Fjord, Nunavut). We see northern Umimmattooq as our place. The place names are ours, because we are the only ones who used to go there. We used to hunt narwhal, walrus, polar bear, and muskox there. But now Canada enforces the border, even with a helicopter, so no one from Qaanaaq wants to risk getting stopped. The border feels artificial to us. Some of our ancestors came from the Canadian side, and we have relatives there. I visited Aujuittuq, Mittimatalik (the Inuktitut name for the community of Pond Inlet), and Iqaluit when I was young, to find our relatives with my mother. One time, Japanese tourists came to Siorapaluk, a settlement just north of Qaanaaq, where we were then living, and we were able to join their flight to Aujuittuq for free. Today, we need passports to go to Canada, which many of us do not have, because we cannot get them in Qaanaaq. And it can be very expensive to travel anywhere. The discussions about creating Pikialasorsuaq (an Inuit-designated protected area in the northern part of Baffin Bay, between Canada and Greenland) may create an opportunity for us to travel freely again to Umimmattooq. Apparently, one sticking point in the negotiations between Canada and Greenland concerns who would be eligible to cross the border without formalities. Canada wants to limit it to Inuit, whereas Greenland says everyone who lives here is a Greenlander and should be treated the same. I just know that I want to be able to do what we used to do, without restrictions.

### 4.2. Travel and Sharing, by Robert Comeau

The tides do not care about boundaries on a map. Nor do the prevailing currents or dominant winds. The animals we rely on consistently cross national lines through the ocean. Most importantly, we as people created our own spaces based around the ocean.

Every summer I get to travel across Baffin Bay on an expedition cruise ship. It is a privilege to be able to travel with other Inuit to the other side of the bay to Kalaallit Nunaat. What an experience to see hunters, seamstresses, business folks that are so similar to us! Their dialect may be a bit different but it is not hard to find commonalities. The welcome that we get as Inuit who come from Canada is the same that any Greenlander would get in Canada. One of warmth and hospitality.

So often times when getting to meet people all over the Greenland coast, we invite them to come visit us in our homeland. But there is a lack of direct flights from Greenland to Nunavut. Needing a passport also makes it more difficult for folks to travel inside Inuit Nunaat.

The new efforts put in by Inuit to manage our waters are just another example of our relationship not only to the water but to each other. These efforts could have an effect on how cruise ships operate.

Mechanisms like Tallurutiup Imanga, a marine protected area in Nunavut, or Pikialasorsuaq, the proposed Inuit-managed area in northern Baffin Bay, could have positive consequences on connecting Inuit. The Pikilarsorsuaq Commission has three recommendations and one of them is a free travel zone between the communities that depend on this polynya, a regularly occurring area of open water within sea ice known also as the North Water. A cruise ship offers a unique opportunity to access the waters that separate Greenland and Nunavut. Imagine our using cruise ships as one of the ways to travel to each other again.

This could lead to so many neat collaborations through the arts, hunting, music, and governance. For myself, it is the qajaq (commonly spelled "kayak" in English) and umiaq (larger, open skin boat) that are tremendous examples of how Inuit have maintained connections to each other over large

bodies of water. Qitdlarssuaq, an Inuq from Baffin Island, shared the knowledge of the qajaq with Avanersuamiut, the people of northern Greenland, in the 19th century when they did not have the practice. Now, Avanersuamiut are essential to Inuit in other regions who are looking to regain the ability to build and hunt on the ocean by qajait (plural of qajaq).

## 5. Discussion

Different groups of Inuit and Chukchi have always had boundaries and territories [23], but the sea has always been an avenue of travel and trade as well as the source of food, clothing, materials, and other resources necessary for life in the Arctic [3,6]. Historically, the arrival of Europeans, as explorers, traders, and whalers, and later as government agents and enforcement personnel, first influenced Inuit and Chukchi practices, then subverted them [6,14], and eventually changed or barred them entirely when it came to crossing the newly created international borders.

In recent decades, Indigenous organizations and governments have fought to restore at least some of the traditional patterns of mobility, interaction, trade, intermarriage, and shared responsibility for a shared environment [24]. In some cases, these efforts have been part of the policies and practices of national governments, such as the visa-free travel program between Alaska and Chukotka mentioned by Vera Metcalf, and the inclusion of Iñupiaq, Yupik, and Chukchi in the Russia-U.S. polar bear treaty [25]. In other cases, Inuit have acted where governments have done little or nothing, such as the Iñupiat-Inuvialuit agreements to manage polar bears and other species in the Beaufort Sea [26]. And in yet other cases, Inuit have carried on or wish to carry on in the face of government opposition, as described above by Carla SimsKayotuk and Toku Oshima.

The six narratives presented here also illustrate competing trends of Indigenous activism and leadership to restore connections and rights, versus government efforts to tighten requirements for documentation and control. Vera Metcalf, Eduard Zdor, Richard Binder Sr., and Robert Comeau celebrate connections and cooperation. Carla SimsKayotuk and Toku Oshima decry recent and forthcoming restrictions that separate families and block traditional patterns of land use. Whether the Bering Strait boundary between Russia and the U.S. remains relatively open, or whether the U.S–Canada and Canada–Greenland borders continue to tighten, remains to be seen.

These transboundary efforts have received relatively little academic attention, especially across the geographic range from Chukotka to Greenland. The perspectives included in this paper are introductory examples of the lived experiences of those living near recently imposed national borders. To gain a more complete picture of Indigenous transboundary experiences in the region, much more can and should be done to document these experiences more fully, from a wider range of individuals involved in the many issues that connect and separate communities on both sides of a border. These issues include wildlife management, trade, family ties, human rights, traditional practices, Indigenous languages, and more. The stories in this paper have only touched on some of these topics, but are nonetheless a first attempt to bring together experiences from across the entire region and all three of today's borders. Creating places for more people to tell their stories will add to the richness of the experience on record, providing a more thorough account of what it is like to live near a national border that separates Indigenous homelands and families. More stories will also add depth and detail to the ways in which connections and interactions across borders are strengthening, largely through Indigenous leadership, and the ways in which those ties are being further harmed by government restrictions.

The cultural unity of the region is in sharp contrast to its territorial division by Russia, the U.S., Canada, and Denmark. The colonial history within the boundaries of each country has affected Inuit and Chukchi through assimilation, cultural and linguistic repression, and enforcement of legal policies and practices regarding travel, hunting, and other aspects of traditional and modern life [2,9]. Thus, today there are newfound differences even among families whose histories placed them on opposite sides of national borders. These differences, however, have proved relatively minor in comparison with the shared desire to re-establish ties and take back the leading role in determining what happens to the peoples, the lands, and the waters of the Arctic. Much remains to be done,

as can be seen in the gap between the aspirations of the Inuit Circumpolar Council (an organization of Inuit from Chuktoka, Alaska, Canada, and Greenland) to achieve Inuit authority for wildlife management [27] and the reality of government control that is the case today within each national jurisdiction. Nonetheless, change starts with vision and commitment, which are demonstrated daily in the actions of Inuit and Chukchi throughout their homeland. Waters that have become barriers are again turning into connectors, helping re-establish traditional Inuit and Chukchi practices alongside modern notions of national sovereignty in the transboundary seas of the North.

**Author Contributions:** Conceptualization, H.P.H.; methodology, H.P.H.; writing—original draft preparation, H.P.H., R.B.S., R.C., V.M., T.O., C.S., E.Z.; writing—review and editing, H.P.H., R.B.S., R.C., L.K.H., V.M., T.O., C.S., E.Z. All authors have read and agreed to the published version of the manuscript.

**Funding:** This research received no external funding.

**Acknowledgments:** We thank Jeremy Davies for preparing Figure 1.

**Conflicts of Interest:** The authors declare no conflict of interest.

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
