# Peer review of "Crossroads of Continents and Modern Boundaries: An Introduction to Inuit and Chukchi Experiences in the Bering Strait, Beaufort Sea, and Baffin Bay"

_water, doi:10.3390/w12061808_

Round 1
Reviewer 1 Report
This article is a very nice introduction to the subject. It's presentation, given the first hand "testimony" is a nice novel way report on the issues.
I am attaching a copy of the article with comments. I have no comments regarding the substance of the article. But, I felt that the authors' familiarity with the subject matter and the region caused them to forget that the reader is not as familiar with the locations and other terms. A little bit of work/effort to respond to my comments will will improve the piece immensely.
There are a few grammatical issues that need attention. I have highlighted these. These are the only comments I have. Besides, there some comments you need be attention as followed:
1. I was unable to check the references cited by the authors, due to the fact the majority of the references were books. As our libraries are closed due to the COVID19 pandemic, I had no access to them.2. The article had many misspellings and grammatical errors. I noted these in my comments on the article.

Author Response
Many thanks for reading our article carefully, for your kind words, and for the corrections to our mistakes. We have made most of the changes you suggest, including grammatical corrections and better explanations of terms and locations. Thank you for helping us improve the manuscript.
Some of the information you asked for (e.g., the location of Umimmattooq, line 194 of the original manuscript) is already provided on the map.
You asked whether the Koryak are the same as the Korak (line 40 of the original manuscript). We are not familiar with the term "Korak," so cannot say for sure. This may be the result of different ways of transliterating Russian words. The Koryak live in the northern part of the Kamchatka Peninsula and the southern part of Chukotka, and this is the way we have always seen the word spelled in Roman letters, if that helps?
You also asked whether we have a reference for the ambiguous status of persons born on one side of the U.S. (line 166 of the original manuscript). We do not, and this lack of attention is part of the need for more stories and information from people living close to the borders described in this paper.
Reviewer 2 Report
Review: Crossroads of Continents and Modern Boundaries: An Introduction to Inuit and Chukchi Experiences in the Bering Strait, Beaufort Sea, and Baffin Bay
This paper is a unique study of the challenges that international borders in the arctic impose on the cultural, social, and economic lives of the indigenous people in the region. The paper focuses on three borders, US-Russia, US-Canada, and US-Greenland, all of which separate peoples that traditionally interacted.
Overall, the paper is fascinating and raises many important issues. However, its contribution to the literature is not well-defined, it does not pose a direct research questions as clearly as would be needed, and its intellectual framework is currently too limited. One specific issue throughout the paper is that the indigenous authors mention rules, policies, or challenges without sufficient context. This context should be provided in the introduction. These issues can be remedied as described below.
Major comments:
- The introduction needs to state a contribution to the literature more clearly. If the difficulties of international borders to indigenous ways of life have been discussed previously, simply stating this, citing relevant literature, and stating that this paper is offering an arctic perspective would be sufficient. If the arctic perspective has already been offered, then the authors would need to state what this paper adds.
- What is the research question? Is there a hypothesis or thesis statement that could be provided early on? Something like: modern international borders continue to constrain the economic, social, and cultural lives of indigenous peoples in the arctic. Then, the perspectives of the indigenous authors would provide qualitative/narrative based evidence in support of the hypothesis/thesis.
- The introduction should do a better job laying out the geography and institutional constraints of the people that each author is affiliated with.
- How is each group’s territory defined within their respective countries? How is their citizenship defined (both their indigenous and national citizenship)? Authors mention lack of passports, split families, rules against movement, bans from hunting, etc. I am an “expert” and still feel lost (I know a lot about mainland US reservation-based rules and policies, but have no context for Inuit governance, and know nothing about Canada, Greenland, or Russia).
- What is the current understanding in the literature of laws governing the borders? These issues should be discussed succinctly prior to the narrative accounts. For instance, take lines 61-65 and expand considerably. Include a paragraph on how each border operates, key agreements, and key challenges to the extent this information is available.
- Greenland in particular requires additional explanation. For one, it is not an independent country cut is discussed as such (not as a part of Denmark) which suggests the governance of indigenous peoples on Greenland might be quite different than for the other three nations. When Toku Oshima writes about a disagreement over travel rules between Greenland and Canada, I felt it would have been helpful to have some context earlier in the paper (I ended up googling Greenland).
- The discussion misses an opportunity to tie the authors’ perspectives together.
- There are clearly some issues facing indigenous arctic peoples relating to tribal sovereignty, the right to free movement, etc. More details on why the Jay Treaty has not been respected by national legislation and reasons why the United Nations Declaration on the Rights of Indigenous Peoples has not been respected by national legislation would be useful in the introduction. It’d be nice to have them for added comprehension of the actions taken by the 4 nations involved.
- If any of the authors of the personal accounts can provide their own insight, that would also be beneficial.
Minor comments
- I would suggest using a polar projection on the map and including the international boundaries more clearly. Russia and Greenland are in parentheses while USA is not.
- There is little distinction made in the introduction/discussion between the impact of historic and colonial policies versus current policies, e.g. lines 48-52.
- The authors themselves seem to be focusing on this work on current international boundary issues, both the removal of severe restrictions (ice curtain) and remaining barriers.
- Lines 255-6 allude to some issues not mentioned explicitly in the paper. “[A]ssimilation, cultural and linguistic repression, and enforcement of legal policies and practices” are a huge set of issues that are not really discussed by the indigenous authors directly.
- Perhaps there is a way to emphasize the focus of the paper while also acknowledging that this paper (or any one paper) cannot provide a full accounting of the links between historic colonial and assimilation practices, the imposition and changing international borders, and current policies.
- Line 48: Europeans should be European
- Quote lines 57-60 needs formatting
- Lines 97-103: I read through this a few times and I still don’t quite understand the geography. Perhaps just a rewrite for clarity.
- Lines 241-44: This may well be true but it’s not entirely consistent with the narrative provided by all the authors. In particular, it seems the imposition of the Soviet-US boundary imposed heavy restrictions that have subsequently reduced, allowing reconnection. However, Toku Oshima seems to describe a border that has become more restrictive recently (he used to hunt, but is no longer able to cross).
- Lines 244-6: “In recent decades, Indigenous organizations and governments have fought to restore at least some of the traditional patterns of mobility, interaction, trade, intermarriage, and shared responsibility for a shared environment” (pg. 6Another sentence on the progress being made on these fronts, or limitations to progress, would be helpful for understanding the current state of these efforts.
- Who are the members of the Inuit Circumpolar Council? Details on what their general aims would be helpful, without describing them completely for brevity’s sake. I believe this Council is mentioned at the end of the discussion and nowhere else.
Author Response
Thank you for a careful reading of our paper and for your kind words, as well as the suggestions for improvements. Here is what we have done, following your numbered points:
- We have added a paragraph in the Introduction (to describe better our contribution to the literature, noting that the topics we address have been raised elsewhere (with three new citations), but not to our knowledge in the Arctic. The new paragraph starts with "Today, Inuit and Chukchi ..." (a sentence from the original manuscript) and then adds more detail.
- We have included our research aim in the new paragraph mentioned above. We did not begin with a hypothesis, but simply with the aim to gather stories from Indigenous peoples living along the three boundaries.
- We have provided more information of this kind in an expanded second paragraph of the Introduction, with citations. However, one could write many volumes on these topics, and we have elected not to go into great detail. The legal status of Indigenous peoples in the four countries is beyond the scope of this paper and we believe would serve to distract from the space and emphasis given to the six narratives. We hope that the additional context material will enhance the value of the narratives.
- Good point. We have added a new paragraph to the Discussion (now the third paragraph) in an attempt to address this point and better connect the competing themes of cooperation and separation that are found in the narratives.
- We agree that tribal sovereignty and related topics are important, but they are largely beyond the scope of this paper. We are not qualified to comment on the long history of recognizing or ignoring obligations such as the Jay Treaty, the UN Declaration on the Rights of Indigenous Peoples, etc. We have added a more recent citation about Arctic governance and cross-border cooperation, for readers interested in learning more.
- We have added the Russia-US and Canada-Greenland maritime borders and made the US-Canada land border thicker. We have made the labels consistent for the four countries. The map is already a polar projection. Map information has been added, along with a graticule, which should help indicate the projection.
- We have added "Historically" at the start of the paragraph that began on line 48 in the original manuscript, to distinguish the description of the past more clearly from the description (and focus) on the present. The material at (former) lines 255-6 is part of the Discussion and has citations, since it is not already covered in the narratives. We provide it to add some context for the idea of cultural unity and cooperation. We have added to the Introduction the comment that no one paper can cover everything (see #3 above).
- Changed
- The formatting was changed from the way we submitted it to the version sent to reviewers. This is a matter for the editor and publisher, and we will draw it to their attention and check the proofs carefully (if the paper is accepted for publication).
- We have tried to improve the description of the geography, a problem also pointed out by Reviewer 1.
- We have added "Historically" at the start of this sentence, too, to make the distinction of past vs. present.
- We have moved this sentence to the following paragraph, which has the details requested, and we added the visa-free travel program between Alaska and Chukotka, both here and in Vera Metcalf's narrative.
- We added a parenthetical description of the Inuit Circumpolar Council.
Round 2
Reviewer 2 Report
I am satisfied with the author's revisions. I look forward to seeing this paper published.